# Identification of FGF13 as a Potential Biomarker and Target for Diagnosis of Impaired Glucose Tolerance

**DOI:** 10.3390/ijms24021807

**Published:** 2023-01-16

**Authors:** Qi Chen, Fangyu Li, Yuanyuan Gao, Fengying Yang, Li Yuan

**Affiliations:** Department of Endocrinology, Union Hospital, Tongji Medical College, Huazhong University of Science and Technology, Wuhan 430022, China

**Keywords:** impaired glucose tolerance, diabetes mellitus, FGF13, comprehensive analysis, biomarker

## Abstract

Early identification of pre-diabetes provides an opportunity for intervention and treatment to delay its progression to type 2 diabetes mellitus (T2DM). We aimed to identify the biomarkers of impaired glucose tolerance (IGT) through bioinformatics analysis. The GSE76896 dataset, including non-diabetic (ND), IGT, and T2DM clinical samples, was deeply analyzed to identify 309 Co-DEGs for IGT and T2DM. Gene ontology (GO) and Kyoto Encyclopedia of Genes and Genomes (KEGG) pathway analyses indicated that inflammatory responses and the PI3K-AKT signaling pathway are important patho-physiological features of IGT and T2DM. Protein–protein interaction (PPI) network analysis and cytoHubba technolgy identified seven hub genes: namely, CCL2, CXCL1, CXCL8, EDN1, FGF13, MMP1, and NGF. The expression and ROC curves of these hub genes were validated using the GSE38642 dataset. Through an immunofluorescence assay, we found that the expression of FGF13 in islets of mice in the HFD and T2DM groups was significantly lower than in the control group. Similarly, the level of FGF13 in the sera of IGT and T2DM patients was lower than that in the healthy group. Together, these results suggest that FGF13 can be treated as a novel biomarker of IGT, which may provide new targets for the diagnosis and treatment of pre-diabetes and T2DM.

## 1. Introduction

Type 2 diabetes mellitus (T2DM) is a major disease threatening global public health, which has caused massive burdens on the health of various populations and economies. It has been reported that the total global number of adults with diabetes was 463 million in 2019, and this number is expected to increase to 700 million by 2045 [1]. Although T2DM has traditionally been considered a disease of the middle-aged and elderly, the epidemic of obesity has led to an increasing number of youths with pre-diabetes and T2DM [2,3]. Of particular concern, patients diagnosed at a younger age appear to have more rapid deterioration in beta-cell function and an increased risk of complications, compared with patients with late-onset T2DM [4,5]. Pre-diabetes is a high-risk metabolic state between normal and diabetic blood glucose homeostasis. Lifestyle or drug interventions can effectively reduce the conversion of prediabetes to diabetes in patients with impaired glucose tolerance (IGT) [6]. Thus, early identification and intervention of prediabetes provides an opportunity to reduce the incidence of diabetes and related complications. 

The development of T2DM occurs through an asymptomatic stage: namely, pre-diabetes. Pre-diabetes has a lifetime risk of up to 70% for conversion to type 2 diabetes, and is associated with increased risks of cardiovascular disease and all-cause mortality [7,8]. Early and accurate diagnosis of pre-diabetes and intensive lifestyle modification in those at high risk are essential. However, pre-diabetes may not be recognized for many years before the progression to diabetes, and a consensus is lacking on the optimal definition of this intermediate metabolic state. The American Diabetes Association has defined pre-diabetes in terms of three distinct conditions: namely, impaired fasting glucose (IFG), impaired glucose tolerance (IGT), and elevated glycosylated hemoglobin levels (5.7–6.4%: IA1c), or a combination of these [9]. At present, fasting plasma glucose (FPG) and HbA1c are commonly used for the diagnosis of T2DM, but their sensitivity in the diagnosis of pre-diabetes is low. Barry et al. [10] have demonstrated that the combination of HbA1c and FPG did not significantly improve diagnostic performance. Although OGTT is the gold standard for the diagnosis of IGT and T2DM, it is not suitable for population-wide screening as it is time-consuming and difficult to repeat. For this reason, there is a need to find a more convenient and reliable biomarker for the diagnosis of pre-diabetes.

It has been shown that the inflammatory response triggered by proinflammatory factors influences the progression of T2DM [11,12]. Serum levels of Il-1β, TNF-α, and IL-6 have been shown to be significantly increased in pre-diabetic patients, compared with normoglycemia [13]. In addition, some studies have observed increased macrophage infiltration in islets of T2DM patients, which is closely related to inflammation and insulin resistance [14,15]. In recent years, metabolomics-based studies have shown that a few metabolites—such as amino acids, free fatty acids, triglycerides, ceramides, and high-density lipoprotein—may be promising biomarkers [16,17,18]. C. Gar et al. [19] demonstrated that elevated branched-chain amino acids and decreased glycine are currently the most stable amino acid markers for prediabetes, insulin resistance, and future type 2 diabetes. In addition, some long non-coding RNAs (lncRNAs) [20,21] and microRNAs (miRNAs or miRs) [22] may also be potential diagnostic biomarkers for prediabetes. Although these results provide the basis for novel pathophysiological hypotheses and treatments for prediabetes, no biomarkers have yet been incorporated into routine screening. The primary research and clinical diagnosis and treatment of pre-diabetes are still progressing and are in an exploratory stage. In this study, we attempt to determine the key genes and potential molecular mechanisms of pre-diabetes through bioinformatics analysis. By collecting the general data and blood samples of normal, pre-diabetes, and diabetes patients, we analyze and compare the expression of key gene regulators, as verified by animal models, and finally provide a basis for finding new biomarkers of pre-diabetes.

## 2. Results

### 2.1. Identification of Differentially Expressed Genes for IGT and T2DM

Gene Expression Omnibus (GEO) is a public database of microarray/gene profiling, from which 55 non-diabetic (ND), 15 IGT, and 55 T2DM pancreatic tissue gene expression profiles of the GSE76896 were obtained. We used the Limma package to process the dataset GSE76896, in order to identify differentially expressed genes (DEGs). Compared with ND samples, 5414 DEGs were identified in IGT samples, including 2859 up-regulated genes and 2555 down-regulated genes. We drew a volcano map (Figure 1A) and a hierarchical clustering heat map of the DEGs (Figure 1B).

The same method was used to analyze the DEGs in the T2DM samples, and we found that there were 2191 DEGs, including 1087 up-regulated genes and 1104 down-regulated genes. The results were mapped as a volcano map (Figure 1C) and a heat map (Figure 1D).

### 2.2. Functional Enrichment Analysis

Next, DEGs from IGT and T2DM samples were intersected to obtain 309 Co-DEGs with 156 common up-regulated genes and 153 common down-regulated genes (Figure 2A). We used the David online tool to perform GO analysis of these 309 Co-DEGs, including molecular functions (MFs), cellular components (CCs), and biological processes (BP); see Figure 2B. The BPs were mainly focused on the inflammatory response, animal organ morphogenesis, cellular response to tumor necrosis factor, neutrophil chemotaxis, and chemotaxis; CCs were mainly focused on the extracellular region, extracellular space, plasma membrane, extracellular matrix, and cell surface; and MFs were mainly focused on cytokine activity, extracellular matrix structural constituent, retinol dehydrogenase activity, chemokine activity, and bile acid binding. We then performed KEGG enrichment analysis using the KOBAS website and identified 14 KEGG pathways (Figure 2C). In addition, the KEGG pathway analysis showed that the DEGs were closely related to the PI3K-Akt signaling pathways, pathways in cancer, and metabolic pathways.

### 2.3. PPI Network and Hub Gene Identification

We submitted Co-DEGs to the STRING for protein–protein interaction (PPI) analysis, resulting in a total of 235 nodes and 292 edges, and used the Cytoscape software for PPI network visualization (Figure 3). In the figure, pink indicates up-regulated genes, while blue indicates down-regulated genes. Then, we used three algorithms—namely, Maximal Clique Centrality (MCC) (Figure 4A), Degree (Figure 4B), and Maximum Neighborhood Component (MNC) (Figure 4C)—to identify the top 10 hub genes. Hub genes play a critical role in biological processes, and the regulation of other genes in related pathways is often influenced by these genes. Finally, we intersected the key genes of the three algorithms to obtain a total of 7 hub genes; namely CCL2, CXCL1, CXCL8, EDN1, FGF13, MMP1, and NGF (see Figure 4D).

### 2.4. Validation of Hub Genes

To verify the screened genes, we analyzed the expression of hub genes in the pancreatic tissue gene expression profile of the GSE38642 dataset (including 9 ND and 9 T2DM). As shown in Figure 5A,B, the expression of CCL2, CXCL1, and FGF13 was significantly down-regulated (all *p* < 0.05) in the islets of T2DM, compared to the control group, in datasets GSE76896 and GSE38642. MMP1 was down-regulated in GSE76896, but up-regulated in GSE38642 (Appendix A). CXCL8, EDN1, and NGF were down-regulated in GSE76896 and not significantly different in GSE38642 (Appendix A). Furthermore, we plotted the ROC curves of hub genes and calculated the area under the curve (AUC) to distinguish T2DM from the control group. The diagnostic values of hub genes in GSE76896 were as follows: CXCL1 (AUC, 0.631), FGF13 (AUC, 0.62), and CCL2 (AUC, 0.6); see Figure 5C. In addition, they also had high diagnostic values in the GSE38642 dataset (Figure 5D). CCL2 and CXCL1 are common chemokines that play a role in a variety of diseases. As an emerging biomarker, changes in FGF13 in patients with IGT or T2DM have not been investigated, so we chose FGF13 for further study.

### 2.5. FGF13 Was Down-Regulated in Islets of Mouse Models of Prediabetes and T2DM

Through bioinformatics methods, we found that FGF13 was a down-regulated DEG in islet samples of IGT and T2DM, being not only highly correlated with pathways obtained by differential gene enrichment analysis, but also serving as a hub gene closely related to the three algorithms in the PPI network of the DEGs. We further verified the changes in FGF13 in the islets of pre-diabetes and type 2 diabetes mouse models (Appendix A). We observed a significant increase in body weight in HFD-fed mice with or without STZ administration (Appendix A). Fasting plasma glucose was significantly higher in the HFD and T2DM groups than in the SD group (Appendix A). As expected, these mice showed marked impaired glucose tolerance and insulin resistance (Appendix A). In addition, the number of islet α-cells increased in the HFD group, and more significantly in the T2DM group, while the α/β-cell ratio was dysregulated (Appendix A). Interestingly, the immunofluorescence results indicated that islet FGF13 expression was down-regulated in the high-fat group mice, compared with the SD group, and further down-regulated in the T2DM group (Figure 6A,B).

### 2.6. Verification of FGF13 in Clinical Samples

In order to verify the clinical application potential of the FGF13 gene, the ELISA method was used to detect the protein level encoded by the FGF13 gene in clinical samples. The blood glucose level in the IGT and T2DM groups was higher than that of the control group, and the difference was statistically significant (Table 1). The expression of serum FGF13 in the IGT and T2DM groups was lower than that in the control group, and the difference was also statistically significant (Figure 7). These findings indicate that the identified FGF13 is a potential biomarker for the early diagnosis and prognostic prediction of IGT.

## 3. Discussion

Early identification and intervention of pre-diabetes provide an opportunity to reduce the incidence of diabetes and the incidence of related complications. Therefore, it is necessary to find a convenient and reliable biomarker to diagnose prediabetes. In this study, we obtained the GSE76896 dataset from GEO and identified 309 Co-DEGs of IGT and T2DM by differential expression analysis. GO analysis of these 309 Co-DEGs indicated that these genes are mainly involved in the inflammatory response, neutrophil chemotaxis, and other processes, and are enriched in Metabolic pathways, Pathways in cancer, and the PI3K-Akt signaling pathway. Then, based on the PPI network and cytoHubba technology, we identified seven hub genes, namely, CCL2, CXCL1, CXCL8, EDN1, FGF13, MMP1, and NGF. Next, we used the GSE38642 dataset to validate the expression and diagnostic value of these hub genes. We finally identified CCL2, CXCL1, and FGF13 as the most diagnostic hub genes in both GSE76896 and GSE38642 datasets.

To date, there have been some reports focused on hub genes and T2DM. We analyzed these seven genes in the next step. CCL2, CXCL1, and CXCL8 are all family members related to chemokines. Chemokines, as chemoattractants for various immune cells, have increasingly been reported to be involved in the occurrence and development of metabolic diseases [23,24,25]. The chemokine CCL2 has been identified as monocyte chemoattractant protein 1 (MCP1) or small inducible cytokine A2, and its primary function is to recruit monocytes to inflammatory tissues [26]. The expression level of CCL2 in adipocytes of obese individuals has been found to be significantly increased, and the expression of inflammatory factors TNFα and IL-6 in adipose tissue was significantly correlated with CCL2 expression [27,28]. Recent studies have shown that higher CCL2 concentrations are associated with the risks of early IR and the development of obesity. In addition, CCL2 can interact with its specific receptor CCR2, playing a primary role in regulating the islet immune microenvironment and monocyte infiltration and migration [29]. Inhibition of CCL2 improves glucose disorder, significantly reduces the progression of diabetes complications, and improves the survival rate of islet transplantation in patients with T1DM [26,30,31,32,33]. The chemokine CXCL1 is a catalyst for neutrophils, and the CXCL1/CXCR2 axis plays a crucial role in recruiting neutrophils to respond to microbial infection and tissue damage [34]. Increased CXCL-1 levels have been linked with obesity, hyperglycemia, and myocardial infarction in patients. Studies have shown that CXCL1 is involved in the pathophysiological process of T1DM. CXCL1 levels are elevated in the blood of patients with T1DM [25] and have been correlated with the rate of disease progression [35]. Neutrophil accumulation in the pancreas has been demonstrated in patients with T1DM [36], which increases islet destruction in the early stages of T1DM. In addition, CXCL1 levels are also elevated in patients with T2DM and gestational diabetes mellitus, leading to decreased insulin secretion and degraded islet function [37,38]. However, Rebuffat et al. [39] have shown that CXCL1 levels were highly expressed in rat models of PDM, but did not play a direct role in islet β-cell dysfunction. The chemokine CXCL8, also known as IL-8 (interleukin-8), is known to have a strong affinity with CXCR1 and CXCR2 [40]. Over-production of IL-8 may be a key driver of obesity-related chronic inflammation [41]. Clinical studies have shown that CXCL8 secreted by adipocytes may be related to complications such as T2DM [42]. Inhibition of the CXCL8/CXCR1 axis can protect db/db mice from metabolic diseases by regulating inflammation, as well as reducing the inflammation and apoptosis of diabetic renal tubular cells induced by high glucose [43,44]. Islet endothelial cell dysfunction can contribute to β-cell failure, leading to the progression of diabetes. Endothelin-1 (EDN1)—a 21-amino acid peptide expressed by various cell types—can regulate vasomotor tone and vascular remodeling, and is an important marker of endothelial dysfunction [45]. Animal studies have shown that a high-fat diet led to increased islet Edn1 levels in mice [46], whereas blocking endothelin-1 type B receptor improved glucose intolerance and insulin resistance in mice [47]. In addition, NGF has been reported to antagonize molecular alterations downstream of insulin resistance receptors, suggesting new potential therapeutic targets to slow diabetes-related brain insulin resistance [48].

Fibroblast growth factor 13 (FGF13) is a member of the FGF homologous factor (FHF) sub-family, which is widely expressed in the developmental and adult nervous systems. FGF13 encodes an intracellular protein involved in microtubule stabilization and regulation of the function of voltage-gated sodium channels (VGSCs), and has been identified as a microtubule-stabilizing protein regulating neuronal polarization and migration during neuronal development [49,50]. The abnormal behavior of FGF13-deficient mice suggested that FGF13 plays a vital role in regulating emotional behavior, as well as learning and memory [51]. In addition, FGF13-mediated microtubule stability is required to promote nerve repair after peripheral nerve injury, highlighting the potential therapeutic value of FGF13 in promoting the repair of injured nerves [52]. In addition to the regulation of sodium channels, FGF13 can also regulate other voltage-gated ion channels [53,54] and affects many other physiological and pathological processes, including the regulation of arrhythmia [53], cancer [55], polycystic ovary syndrome [56], smooth muscle and skeletal muscle cell development [57], and inflammatory pain [58]. Recent studies have identified FGF13 as a new candidate gene for obesity [59], which regulates thermogenesis and energy balance. FGF13 heterozygous knockout mice presented impairment of the central nervous system’s sympathetic activation of brown fat, leading to obesity [60].

As an emerging biomarker, changes in FGF13 in patients with IGT or T2DM have not been investigated. In this study, by differential expression analysis, we found that FGF13 is a down-regulated DEG in islet samples of IGT and T2DM, which was not only highly correlated with the pathways obtained by differential gene enrichment analysis, but was also a hub gene closely related to the three algorithms in the PPI network of the DEGs. To confirm the dysregulation of FGF13 in diabetes, we examined FGF13 expression in the endocrine islets of IGT and diabetic mouse models. Islet cells regulate glucose homeostasis by secreting various hormones, and the imbalance and dysfunction of the islet α/β-cell ratio can lead to abnormal glucose tolerance. Consistent with the results of our bioinformatics analysis, we found that FGF13 expression was down-regulated in islets of both prediabetic and diabetic animal models, where this down-regulation was more pronounced in diabetic mice. In order to verify the potential of FGF13 in clinical applications, serum FGF13 protein levels in IGT and T2DM patients were detected by ELISA, and the involvement of FGF13 in the pathogenesis of diabetes was confirmed. Therefore, we believe that the detection of FGF13 levels will be helpful for the diagnosis of IGT, which is also the clinical significance of this study.

The main limitation of this study was the small sample size, which may not allow the results to be fully generalizable. In future work, we intend to further expand the sample size for testing. In addition, the molecular mechanisms underlying the downregulation of FGF13 expression in IGT and diabetes, as well as its role in various metabolic tissues, need to be further explored. In conclusion, this study provides new biomarkers for the diagnosis and prognosis of IGT, as well as new research directions for the prevention of type 2 diabetes.

## 4. Materials and Methods

### 4.1. Data Selection

We obtained the GSE76896 dataset through the NCBI Gene Expression Omnibus (GEO) public database (http://www.ncbi.nlm.nih.gov/geo/, accessed on 1 October 2022). The dataset contains pancreatic tissue samples from 55 non-diabetic (ND) patients, 55 patients with T2DM, and 15 patients with impaired glucose tolerance (IGT). Microarray data were based on the GPL570 (HG-U133_Plus_2) Affymetrix Human Genome U133 Plus 2.0 Array (Affymetrix, Santa Clara, CA, USA).

### 4.2. Data Processing

We used the “limma” package in the R software to screen for DEGs between the ND and IGT groups, and between the ND and T2DM groups, respectively, with parameters set to adj. *p* Value < 0.05 and |log_2_FC| ≥ 0.5. Next, we used the “ggplot2” and “heat map” packages to draw volcano maps and heat maps. DEGs with log_2_FC > 0 were considered to be up-regulated genes, while the down-regulated genes were screened according to log_2_FC < 0. The two groups of DEGs were intersected to draw a Venn diagram, obtaining 309 Co-DEGs, with 156 up-regulated genes and 153 down-regulated genes.

### 4.3. Enrichment Analysis

Database for Annotation, Visualization, and Integrated Discovery (DAVID; https://davi.ncifcrf.gov (accessed on 20 October 2022), and KEGG Orthology Based Annotation System (KOBAS; http://kobas.cbi.pku.edu.cn/, accessed on 20 October 2022) were used for functional and pathway enrichment analyses. Gene Ontology (GO) enrichment analysis was carried out on the webpage using the DAVID online tools, including molecular functions (MFs), cellular components (CCs), and biological processes (BPs). KEGG was used for pathway enrichment analyses. Among them, in GO enrichment analysis, *p* < 0.05 was taken as the selection criterion, while the KEGG pathway analysis was based on *p* < 0.01 for selection.

### 4.4. Construction of PPI Network

To further investigate the biological functions of Co-DEGs, a protein–protein interaction (PPI) network was constructed using the STRING tool (also known as PPI network functional enrichment analysis; https://string-db.org/ (accessed on 25 October 2022). Nodes and edges in the PPI network represent the proteins and the interactions between two proteins, respectively. By determining the correlation between proteins through the edges between nodes, we understand the relationship between genes and obtain the hub regulatory genes. We imported the Co-DEGs interaction results into the Cytoscape software for PPI network visualization, and used the cytoHubba plug-in to screen the hub genes. Hub genes play a critical role in biological processes, and the regulation of other genes in related pathways is often influenced by these genes. Therefore, hub genes are often important targets and hot spots for research. We used three algorithms—namely, Maximal Clique Centrality (MCC), Maximum Neighborhood Component (MNC), and Degree—to identify the top 10 hub genes. Finally, the results of the three algorithms were intersected to obtain the final hub genes.

### 4.5. Validation of Hub Genes

To validate the difference in hub gene expression between T2DM and ND, the pancreatic tissue gene expression profile of the GSE38642 dataset was used for validation. Microarray data were based on the GPL6244 (HuGene-1_0-st) Affymetrix Human Gene 1.0 ST Array. Nine T2DM patients and nine ND patients were selected from the GSE38642 dataset, and the expression difference in hub genes in GSE38642 was depicted as a boxplot using the draw_boxplot function of the online tool BioLadder (https://www.bioladder.cn/, accessed on 24 December 2022). We also used BioLadder to calculate ROC curves for selected genes, in order to assess the ability of hub genes to diagnose T2DM.

### 4.6. Clinical Patient Selection

The study included 25 patients with T2DM, 15 patients with IGT (7.8 < FBG < 11.1 mmol/L after 2 h of OGTT), and 13 healthy people (FBG < 6.1 mmol/L), selected from the Department of Endocrinology, Union Hospital, Tongji Medical College, Huazhong University of Science and Technology. Exclusion criteria were: 1. Patients with severe systemic diseases, such as moderate to severe renal insufficiency (eGFR < 45 mL/min/1.73 m^2^), severe hepatic insufficiency, decompensated heart failure, myocardial infarction, malignant tumor, severe infection, and so on. 2. Type 1 diabetes mellitus, specific type diabetes mellitus, and insulin-dependent diabetes mellitus. 3. Previous history of pancreatitis. 4. Patients with Cushing syndrome and patients who had previously used glucocorticoids. 5. Patients who have had gastrointestinal or other abdominal operations within the last year. This study was conducted according to the principles of the Declaration of Helsinki, and was approved by the local ethics committee of Tongji Medical College, Huazhong University of Science and Technology, Hubei Province, China (ChiCTR2000034751). All subjects gave their informed consent for inclusion before they participated in the study.

### 4.7. Enzyme-Linked Immunosorbent Assay (ELISA)

The blood samples were collected, and FGF13 levels were determined by using the human FGF13 ELISA kit (HM11488, Bio-Swamp, Wuhan, China) according to the manufacturer’s instructions. Briefly, 40 µL of serum sample was used, and all samples were run in duplicate. Samples were thawed and then incubated with Biotinylated anti-FGF13 antibody and HRP-Conjugate reagent on a microwell plate for 30 min, followed by TMB peroxides for 10 min. Reactions were terminated by 0.25 mol/L of sulfuric acid. These assays were performed at room temperature with five wash cycles between steps. The absorbance at 450 nm was detected using a microplate reader (PerkinElmer, Waltham, MA, USA).

### 4.8. IGT and T2DM Mouse Model

Six-week-old male C57BL/6J wild type (WT) mice were randomly divided into three groups (*n* = 5–6 mice/group): (1) Standard diet control group (SD); (2) High-fat diet group (HFD); and (3) T2DM group. Mice in the SD group were fed a standard chow diet. The HFD mice were fed a HFD (#D12492) for 16 weeks. In the T2DM group, mice were first fed a HFD for 4 weeks, then injected with STZ (30 mg/kg, Sigma-Aldrich, St. Louis, MO, USA) for 3 consecutive days to induce T2DM, as previously described [61]. The mice with blood glucose levels over 16.7 mmol/L, as measured by a blood glucose meter (LifeScan), were considered as T2DM and were continually fed a HFD until 16 weeks.

All mice were housed in a specific pathogen-free (SPF) animal laboratory, with a 12 h light:12 h darkness cycle and at room temperature (20–22 °C). Their body weight and fasting blood glucose were measured weekly. All experiments were performed according to procedures approved by the Animal Research Committee of Tongji Medical College, Huazhong University of Science and Technology, Hubei Province, China (Reference Number: 2739, Date Approved: 10 October 2020).

### 4.9. Blood Glucose Levels, Intraperitoneal Insulin Tolerance Tests, and Intraperitoneal Glucose Tolerance Tests

Tail fasting blood glucose was measured using a blood glucose meter (LifeScan) after mice had fasted for 12 h. The intraperitoneal glucose tolerance test was performed by injecting glucose (2 g/kg body weight, i.p.) into mice which had fasted for 12 h. Blood was collected from the tip of the tail vein, measured at 0, 30, 60, 90, and 120 min. The intraperitoneal insulin tolerance test was performed by injecting insulin (0.75 U/kg body weight, i.p.) into mice which had fasted for 12 h. Blood was collected from the tail vein’s tip, measured at 0, 15, 30, 60, and 90 min. The area under the curve was calculated using the GraphPad Prism 7.00 software (GraphPad Software, Inc., San Diego, CA, USA).

### 4.10. Immunofluorescence Staining

Paraffin sections of liver tissues were deparaffinized with xylene and dehydrated in a series of 100, 95, 85, and 70% ethanol. The sections were incubated with primary antibodies overnight at 4 °C. After washing three times in PBS for 10 min, the sections were incubated with an anti-rabbit FITC-conjugated (FITC, CY3) secondary antibody (1:100 dilution) for 1 h, and nuclei were stained with DAPI (Servicebio, Wuhan, China) at room temperature for 10 min in the dark [62]. The sections were observed by confocal microscopy (Nikon, Tokyo, Japan). The antibodies used for immunofluorescence are summarized in Table 2.

### 4.11. Statistical Analysis

The data were analyzed using the GraphPad Prism software (7.0.0). Differences in numerical parameters between two groups were assessed by an unpaired two-tailed *t*-test, while a one-way ANOVA was conducted to compare multiple groups. All data are expressed as the mean ± SD, and *p* < 0.05 was considered statistically significant.

## Figures and Tables

**Figure 1 ijms-24-01807-f001:**
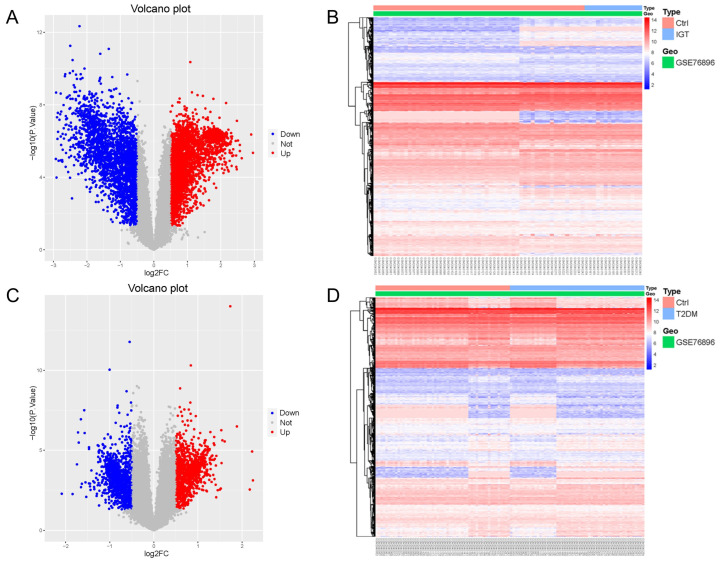
Volcano map (**A**,**C**) and heat map (**B**,**D**) of DEGs for IGT and T2DM.

**Figure 2 ijms-24-01807-f002:**
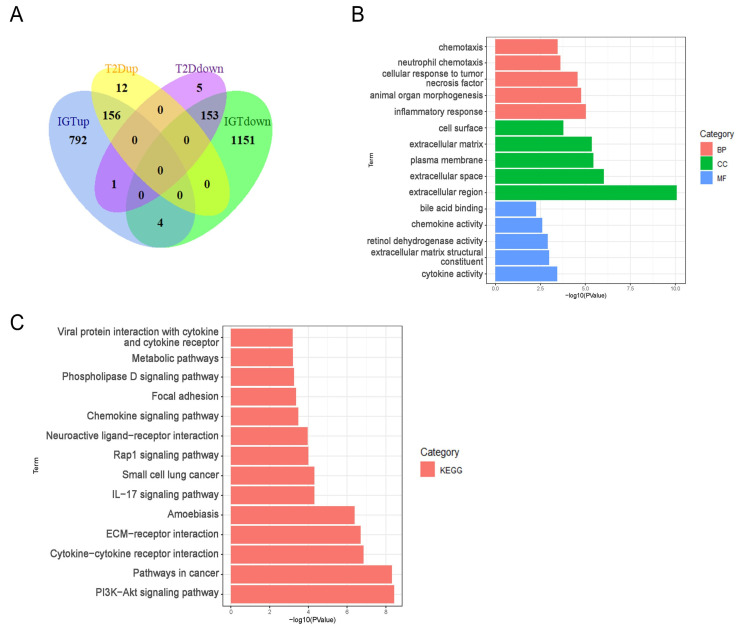
The results of Co-DEGs (**A**); GO of Co-DEGs (**B**); and KEGG of Co-DEGs (**C**).

**Figure 3 ijms-24-01807-f003:**
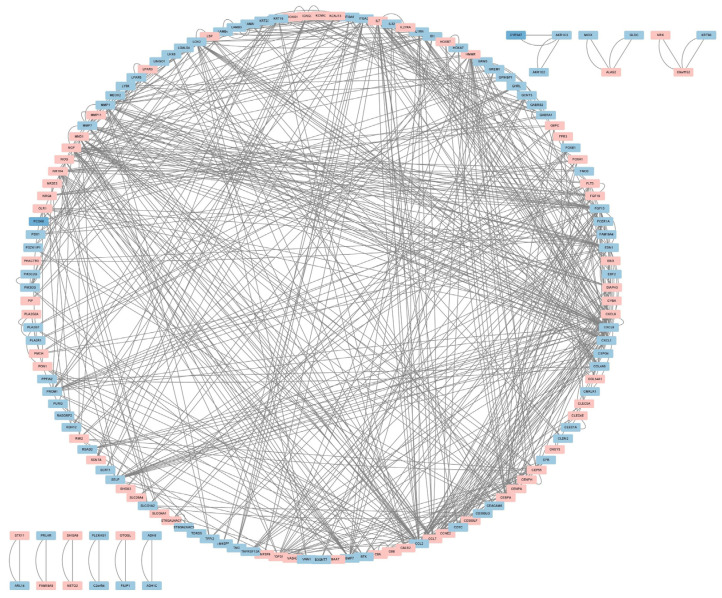
PPI network of the Co-DEGs.

**Figure 4 ijms-24-01807-f004:**
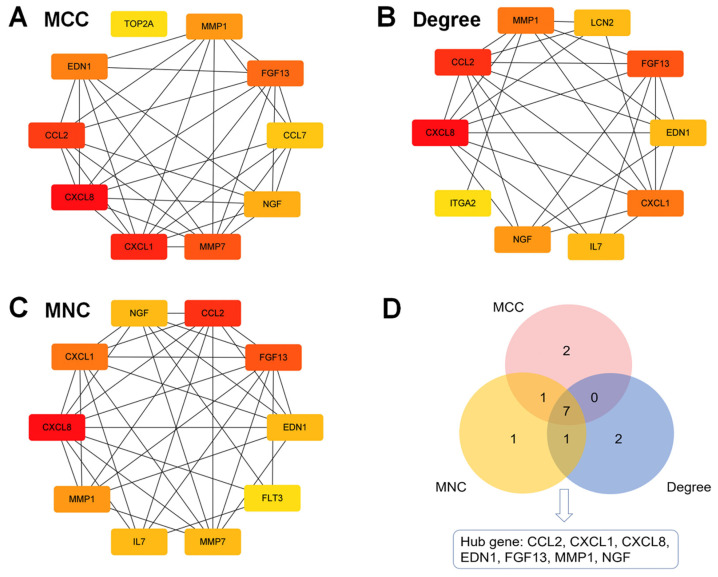
Hub genes in the PPI network by DCC (**A**), Degree (**B**), and MNC (**C**). The Venn diagram illustrates seven hub genes screened by three algorithms (**D**).

**Figure 5 ijms-24-01807-f005:**
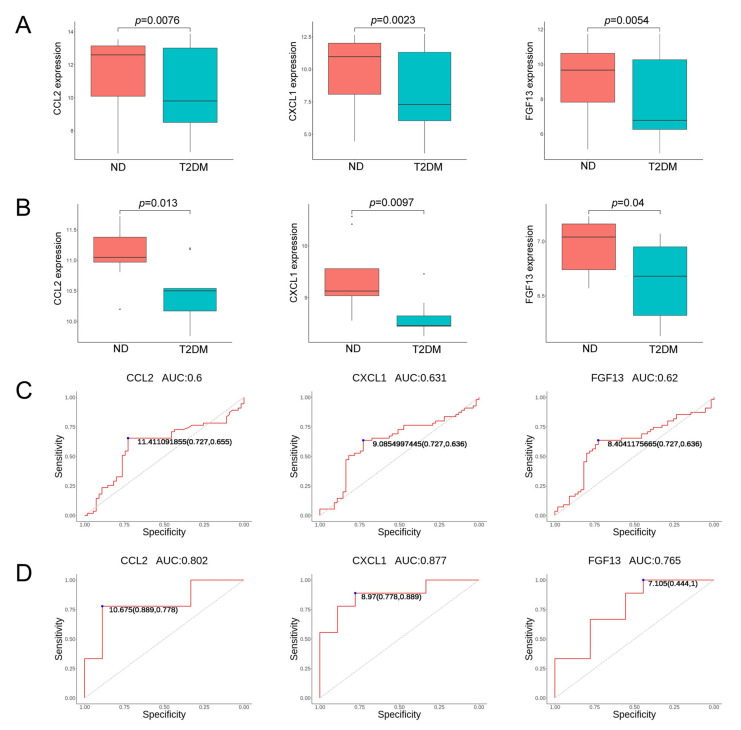
Validation of hub genes: (**A**,**B**) The expression of hub genes (including CCL2, CXCL1, and FGF13) detected in GSE76896 and GSE38642; and (**C**,**D**) ROC curves of the selected genes in GSE76896 and GSE38642.

**Figure 6 ijms-24-01807-f006:**
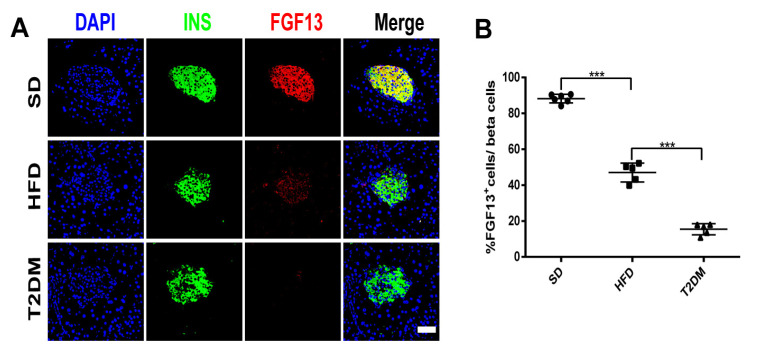
The expression levels of FGF13 in islets of mouse models of pre-diabetes and T2DM. (**A**,**B**) Representative immunofluorescent images of insulin and FGF13 in the pancreatic islets (**A**) and the ratio of double immunopositive cells (**B**). Scale bars = 20 μm. Values are expressed as means ± SD (*n* = 5 or 6). *** *p*  <  0.001.

**Figure 7 ijms-24-01807-f007:**
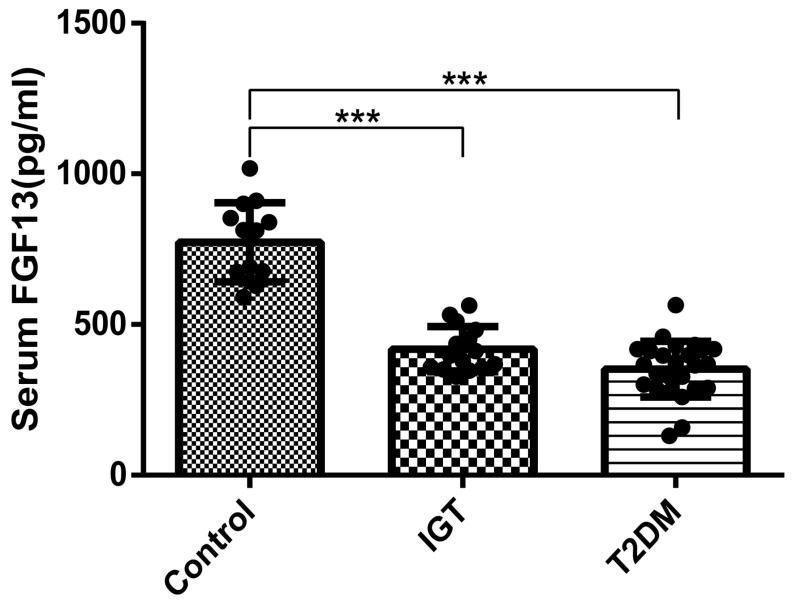
Comparison of serum FGF13 levels among IGT patients, T2DM patients, and control (*** *p* < 0.001 vs. control).

**Table 1 ijms-24-01807-t001:** Comparison of data of patients in three groups (* *p* < 0.05 vs. control).

Indicator (Median)	Control (*n* = 13)	IGT (*n* = 15)	T2DM (*n* = 25)
Age (years)	36	45	51 *
Blood glucose (mmol/L)	4.9	6.7 *	9.5 *
FGF13 (pg/mL)	773.06 ± 125.87	417.92 ± 72.07 *	354.50 ± 92.09 *

**Table 2 ijms-24-01807-t002:** Antibodies for immunodetection.

Antibody	Source	Dilution
Guinea pig anti-insulin	Ab7842 (Abcam, Cambridge, UK)	IF: 1:100
Rabbit anti-glucagon	Ab92517 (Abcam, Cambridge, UK)	IF: 1:100
Rabbit anti-FGF13	26235-1-AP(Proteintech, Wuhan, China)	IF: 1:100

## Data Availability

The datasets presented in this study can be found in online repositories. The names of the repository/repositories and accession number(s) can be found in the article.

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
