# Peer review of "Identification of FGF13 as a Potential Biomarker and Target for Diagnosis of Impaired Glucose Tolerance"

_ijms, 2023, doi:10.3390/ijms24021807_

Round 1

Reviewer 1 Report

this paper is really important for the topic prevention of complication on diabetes and CVR

well written and relevant methodolgy is used

may be could be possibile explain in terms of epidemiology the potential usage of identidication of FGF13 in clinical practice 

Author Response

Response to Reviewer 1 Comments 

Pointthis paper is really important for the topic prevention of complication on diabetes and CVR well written and relevant methodolgy is used

may be could be possibile explain in terms of epidemiology the potential usage of identidication of FGF13 in clinical practice 

Response : Thank you very much for your affirmation and support of our article.

Reviewer 2 Report

The study by Chen et al. identified lower levels of FGF13 in the pancreas and serum of pre-diabetic patients, as well as in mice models of diabetes. The study design is overall sound, and the evidence is convincing. However, the article needs a thorough editing for language before publication can be considered, as multiple errors in grammar, style and vocabulary occur in every paragraph (and even in the title). Additionally, there is repetition in some sentences, as well as too much redundancy/repetition between the sections. If/when all these points are corrected, I'll be able to give more focused comments.

Author Response

Response to Reviewer 2 Comments

PointThe study by Chen et al. identified lower levels of FGF13 in the pancreas and serum of pre-diabetic patients, as well as in mice models of diabetes. The study design is overall sound, and the evidence is convincing. However, the article needs a thorough editing for language before publication can be considered, as multiple errors in grammar, style and vocabulary occur in every paragraph (and even in the title). Additionally, there is repetition in some sentences, as well as too much redundancy/repetition between the sections. If/when all these points are corrected, I'll be able to give more focused comments.

Response: Thank you very much for the constructive suggestion. We feel very sorry for our mistakes. We have checked the manuscript carefully with the help of an English editing service and corrected some tense, grammar, and other errors in the manuscript. In addition, we have corrected duplicated or redundant parts of the manuscript. The changes are shown in red. 

Reviewer 3 Report

In this article, Chen et al. aimed to identify biomarkers for impaired glucose tolerance. They analyzed a dataset from the publicly available database (GEO). Upon differential expression analysis and further pathway analysis, they identified that the differentially expressed genes belonged to inflammatory response, cytokine activity, and chemokine activity pathways. Upon PPI network analysis, they found that CCL2, CXCL1, CXCL8, EDN1, FGF13, MMP1, and NGF were critical “hub” genes. They experimentally demonstrated that among these genes, FGF13 was downregulated in mouse models of T2D as well as in individuals with impaired glucose tolerance and Type 2 Diabetes.

Although these are interesting and novel findings, the study has several caveats that need to be addressed:

 Major:

1. Based on the analyses from the dataset, the authors identified FGF13 as a biomarker. However, based on a single dataset concluding that FGF13 should be utilized as a biomarker seems premature. The authors need to analyze at least two more datasets and see if the findings are consistent. This analysis will not only validate the findings but will also enhance the applicability of their results for the diagnoses.

 2. Another major caveat is also that the current study is a mere association based and lacks mechanistic insight. Although the aim of the study is the identification of biomarkers through bioinformatics, it has been seen with so many studies that the in-silico analyses don’t always translate. As the authors mention, due to the limited sample size, the current findings cannot be confidently accepted. To address this concern, the authors need to increase the sample size and/or perform experiments to demonstrate whether alterations in FGF13 levels are a cause or consequence of the disease. This will further improve its utility as a reliable biomarker.

 3. The authors need to provide an in-depth explanation of the bioinformatic analyses and the observations in the methods and the results section. At this stage, the write-up is too technical and short for the reader’s understanding.

 Minor:

 4. The bioinformatics analyses figure (Fig 1-5) can be clustered into one (or two) Figures as it seems like an unnecessary split.

 5. The authors need to explain the PPI network analyses and what exactly HUB genes are.

 6.The targeted selection of FGF13 among the 7 hub genes for further studies is not explained.

 7.The image quality needs significant improvement. The font size of contents on the axes on each sub-figure must be increased.

 8.In Figs 1 and 2, details about the no. of samples in each group within the study are important.

 9. Fig 6A-H is well established in the field. It’s not novel. It should be moved to the supplemental data section as it serves as a good control.

 10. In line 121, the authors mention, “Next, we investigated the role of FGF13 in islet regulation by constructing prediabetic and type 2 diabetic mouse models”. The sentence doesn’t make sense as nothing new was constructed as these are well-established models.

11.In lines 118, 143-151, it should be FGF13 instead of FDF13.

Author Response

Response to Reviewer 3 Comments

Major:

Point 1: Based on the analyses from the dataset, the authors identified FGF13 as a biomarker. However, based on a single dataset concluding that FGF13 should be utilized as a biomarker seems premature. The authors need to analyze at least two more datasets and see if the findings are consistent. This analysis will not only validate the findings but will also enhance the applicability of their results for the diagnoses.

 Response 1: Thank you very much for the valuable suggestion, which will undoubtedly make our article more meaningful. To verify the screened genes, we analyzed the expression of hub genes in the GSE38642 data set. As shown in Figure 5A and B, the expression of CCL2, CXCL1, and FGF13 was significantly down-regulated (all p < 0.05) in the islets of T2DM, compared to the control group, in data sets GSE76896 and GSE38642. MMP1 was down-regulated in GSE76896, but up-regulated in GSE38642 (Supplementary Figure 1). CXCL8, EDN1, and NGF were down-regulated in GSE76896 and not significantly different in GSE38642 (Supplementary Figure 1). Furthermore, we plotted the ROC curves of hub genes and calculated the area under the curve (AUC) to distinguish T2DM from the control group. The diagnostic values of hub genes in GSE76896 were as follows: CXCL1 (AUC, 0.631), FGF13 (AUC, 0.62), and CCL2 (AUC, 0.6); see Figure 5C. In addition, they also had high diagnostic values in the GSE38642 data set (Figure 5D). CCL2 and CXCL1 are common chemokines that play a role in a variety of diseases. Therefore, the joint validation of the two data sets further supports our study of FGF13 as a new biomarker. The changes are shown in red. 

 Point 2: Another major caveat is also that the current study is a mere association based and lacks mechanistic insight. Although the aim of the study is the identification of biomarkers through bioinformatics, it has been seen with so many studies that the in-silico analyses don’t always translate. As the authors mention, due to the limited sample size, the current findings cannot be confidently accepted. To address this concern, the authors need to increase the sample size and/or perform experiments to demonstrate whether alterations in FGF13 levels are a cause or consequence of the disease. This will further improve its utility as a reliable biomarker.

Response 2: Thank you very much for the constructive suggestion. In order to enhance the applicability of the results for the diagnoses. We further verified and confirmed the expression changes of FGF13 in the islets of T2DM and its diagnostic value by using another dataset. We feel sorry that due to time, we could not collect enough samples of the population. We added 5 samples in the diabetes group, 3 in the IGT group, and 3 in the control group and further verified that serum FGF13 expression in the IGT and T2DM groups was lower than that in the control group. The changes are shown in red. 

Point 3: The authors need to provide an in-depth explanation of the bioinformatic analyses and the observations in the methods and the results section. At this stage, the write-up is too technical and short for the reader’s understanding.

 Response 3: Thank you very much for your rigorous suggestions. We feel sorry that we did not provide a complete description of the bioinformatic analyses and the observations in the methods and the results section. We have revised and further elaborated the in-depth explanation of the analytical methods and results of bioinformatics techniques in the Methods and results section so that readers can better understand the significance of the results. The changes are shown in red. 

Minor:

Point 4: The bioinformatics analyses figure (Fig 1-5) can be clustered into one (or two) Figures as it seems like an unnecessary split.

Response 4: Thank you very much for the constructive suggestion. We have gathered Figures 1 and 2 into one figure. Since other bioanalysis diagrams use different techniques and methods, we still consider dividing them into different figures for description.

 Point 5: The authors need to explain the PPI network analyses and what exactly HUB genes are.

Response 5: Thank you very much for your careful suggestions. We feel sorry that we did not provide detailed instructions. Nodes and edges in the protein–protein interaction (PPI) network represent the proteins and the interactions between two proteins, respectively. By determining the correlation between proteins through the edges between nodes, we understand the relationship between genes and obtain the core regulatory genes. Hub genes play a critical role in biological processes, and the regulation of other genes in related pathways is often influenced by these genes. Therefore, hub genes are often important targets and hot spots for research. We have added corresponding explanations for the " PPI network analyses and HUB genes " in the Methods section. The changes are shown in red.

Point 6: The targeted selection of FGF13 among the 7 hub genes for further studies is not explained.

Response 6: Thank you very much for your suggestions. We are sorry that we did not provide a complete description of the targeted selection of FGF13 among the 7 hub genes for further studies. Many studies have confirmed the correlation between six other HUB genes and T2DM, but FGF13 has not been studied as an emerging biomarker in patients with IGT or T2DM. In this study, through bioinformatics methods, we found that FGF13 was a down-regulated DEG in islet samples of IGT and T2DM, being not only highly correlated with pathways obtained by differential gene enrichment analysis, but also serving as a hub gene closely related to the three algorithms in the PPI network of the DEGs. To verify screened genes, we analyzed the expression of hub genes in the GSE38642 dataset. As shown in Figures 5A and B, the expression of CCL2, CXCL1, and FGF13 was significantly down-regulated (all p < 0.05) in the islets of T2DM, compared to the control group, in data sets GSE76896 and GSE38642. Moreover, we plotted the ROC curves of hub genes and calculated the area under the curve (AUC) to detect the diagnostic value of hub genes. We confirmed the high diagnostic value of CXCL1, CCL2, and FGF13 in both data sets (Figure 5C and D). CCL2 and CXCL1 are common chemokines that play a role in various diseases. Based on the above analysis, we choose FGF13 for the next step. We further elaborated on the reasons for choosing FGF13 for further study in chapter 2.4 of the results and the discussion section. Changes are shown in red.

Point 7: The image quality needs significant improvement. The font size of contents on the axes on each sub-figure must be increased.

Response 7: Thank you very much for your careful suggestions. We are sorry for our negligence and We have carefully improved the quality of the images and increased the font size of the axes on each sub-figure.

Point 8: In Figs 1 and 2, details about the no. of samples in each group within the study are important.

Response 8: Thank you very much for your rigorous suggestions. In chapter 4.1 of the Methods section, we have written that this dataset consists of pancreatic tissue samples from 55 non-diabetic (ND) patients, 55 T2DM patients, and 15 patients with impaired glucose tolerance (IGT). In addition, we also included the number of subgroups for the new GSE38642 dataset in chapter 4.5 of the Methods section. For a more precise expression, we have written the sample numbers of each group in the results section. The changes are shown in red.

Point 9: Fig 6A-H is well established in the field. It’s not novel. It should be moved to the supplemental data section as it serves as a good control.

Response 9: Thank you very much for the constructive suggestion. We have moved Fig 6A-H and J to the supplementary data section as Fig S2A-I. The changes are shown in red. 

Point 10: In line 121, the authors mention, “Next, we investigated the role of FGF13 in islet regulation by constructing prediabetic and type 2 diabetic mouse models”. The sentence doesn’t make sense as nothing new was constructed as these are well-established models.

Response 10: Thank you very much for your suggestions. We have more cautiously modified the expression to " We further verified the changes of FGF13 in the islets of pre-diabetes and type 2 diabetes mouse models" . The changes are shown in red. 

Point 11: In lines 118, 143-151, it should be FGF13 instead of FDF13.

Response 11: Thank you very much for your rigorous suggestions. We are sorry for our  carelessness and we have replaced "FGF13" with" FDF13". The changes are shown in red. 

Round 2

Reviewer 2 Report

The authors have conducted a thorough editing for language, as requested. The resulting manuscript has been considerably improved. There are specific issues that need to be addressed prior to publication:

1)In the title:  "by Comprehensive Analysis" is a non-informative phrase and should be omitted or replaced.

2) In the Abstract and elsewhere: the GSE76896 data set (and the other set) have to be properly introduced upon first mention in a section, as the accession number is in itself non-informative. Please explain what is this dataset instead of giving an accession only.

3) Line 17: replace "screened" with "identified"

4) Lines 181-186: it is not necessary to repeat the table values in the text

5) In the Discussion (lines 201-216) there is excessive repetition of Results.

6) Materials and Methods: Data selection lacks description of the validation dataset.

7) ELISA method description (lines 364 on) should be more detailed

8) Supplementary Fig. 2: panel A is unclear. What does WT stand for?

Author Response

Response to Reviewer 2 Comments

 Point 1: In the title:  "by Comprehensive Analysis" is a non-informative phrase and should be omitted or replaced.

 Response: Thank you very much for the valuable suggestion. We have already omitted the phrase "by comprehensive analysis" in the title. The changes are shown in red. 

Point 2:In the Abstract and elsewhere: the GSE76896 data set (and the other set) have to be properly introduced upon first mention in a section, as the accession number is in itself non-informative. Please explain what is this dataset instead of giving an accession only.

Response: Thank you very much for your constructive suggestions. We feel sorry that we did not introduce the GSE76896 dataset properly in a section when first mentioned. We have revised and further elaborated the in-depth explanation of the GSE76896 dataset in the Abstract and Results section. The changes are shown in red.

Point 3:Line 17: replace "screened" with "identified"

Response: Thank you very much for your rigorous suggestions. We have replaced "screened" with " identified". The changes are shown in red. 

Point 4:Lines 181-186: it is not necessary to repeat the table values in the text

Response: Thank you very much for the constructive suggestion. We have removed the table values from the text in this section. The changes are shown in red.

Point 5In the Discussion (lines 201-216) there is excessive repetition of Results.

Response: Thank you very much for your valuable suggestions, which will certainly make our articles more concise. We have revised the Discussion section of the article. The content is as follows: “In this study, we obtained the GSE76896 dataset from GEO and identified 309 Co-DEGs for IGT and T2DM by differential expression analysis. GO analysis of these 309 Co-DEGs indicated that these genes are mainly involved in the inflammatory response, neutrophil chemotaxis, and other processes, and are enriched in Metabolic pathways, Pathways in cancer, and the PI3K-Akt signaling pathway. Then, based on the PPI network and cytoHubba technology, we identified seven hub genes, namely, CCL2, CXCL1, CXCL8, EDN1, FGF13, MMP1 and NGF. Next, we used the GSE38642 dataset to validate the expression and diagnostic value of these hub genes. We finally identified CCL2, CXCL1 and FGF13 as the most diagnostically valuable hub genes in both GSE76896 and GSE38642 datasets.” The changes are shown in red.

Point 6Materials and Methods: Data selection lacks description of the validation dataset.

Response: Thank you very much for your rigorous suggestions. We feel sorry that we did not provide a complete description of validation dataset. To validate the difference in hub gene expression between T2DM and ND, the pancreatic tissue gene expression profile of the GSE38642 dataset was used for validation. Microarray data were based on the GPL6244 [HuGene-1_0-st] Affymetrix Human Gene 1.0 ST Array. We have added corresponding explanations for the validation dataset in the Methods section 4.5. The changes are shown in red.

Point 7ELISA method description (lines 364 on) should be more detailed

Response: Thank you very much for your rigorous suggestions. We are sorry that we did not provide a complete description of the ELISA method. We have added a further description of the ELISA method in the Methods section. The changes are shown in red. 

Point 8Supplementary Fig. 2: panel A is unclear. What does WT stand for?

Response: Thank you very much for your careful suggestions. We feel sorry that we did not provide detailed instructions for WT. The WT stands for wild-type mouse. We have added corresponding explanations for the “WT” in the Methods section. In order to express panel A more clearly, we have replaced the expression of WT with a different group name. The changes are shown in red.

Reviewer 3 Report

The authors have addressed the majority of my comments satisfactorily. The manuscript can be accepted in its present form.

Author Response

Response to Reviewer 3 Comments

 PointThe authors have addressed the majority of my comments satisfactorily. The manuscript can be accepted in its present form.

Response : Thank you again for your valuable and constructive suggestions to improve the quality of our manuscript. These comments are all precious and helpful for improving our article.
